# Regional hypothermia improves gastric microcirculatory oxygenation during hemorrhage in dogs

Richard Truse[1]*, Michael Smyk[1]¤, Jan Schulz[1], Anna Herminghaus[1], Andreas P. M. Weber[2], Tabea Mettler-Altmann[2], Inge Bauer[1], Olaf Picker[1], Christian Vollmer[1]

**1** Department of Anesthesiology, Duesseldorf University Hospital, Duesseldorf, Germany, **2** Institute of Plant Biochemistry, Cluster of Excellence on Plant Sciences (CEPLAS), Heinrich-Heine-University Duesseldorf, Duesseldorf, Germany

¤ Current address: Institute of Diagnostic and Interventional Radiology, Duesseldorf University Hospital, Duesseldorf, Germany; in partial fulfillment of the requirements of the MD thesis of M. Smyk
* Richard.Truse@med.uni-duesseldorf.de

**Data Availability Statement:** All relevant data are within the manuscript and its Supporting Information files.

## Abstract

Mild systemic hypothermia increases gastric mucosal oxygenation ($\mu HbO_2$) during hemorrhagic shock in dogs. In the context of critical blood loss hypothermia might be fatal due to adverse side effects. Selective regional hypothermia might overcome these limitations. The aim of our study was to analyze the effects of regional gastric and oral mucosal hypothermia on $\mu HbO_2$ and perfusion ($\mu flow$). In a cross-over study six anesthetized dogs were subjected to local oral and gastric mucosal hypothermia (34˚C), or maintenance of local normothermia during normovolemia and hemorrhage (-20% blood volume). Macro- and microcirculatory variables were recorded continuously. During normovolemia, local hypothermia increased gastric microcirculatory flow ($\mu flow$) without affecting oxygenation ($\mu HbO_2$) or oral microcirculation. During mild hemorrhagic shock gastric $\mu HbO_2$ decreased from 72±2% to 38±3% in the normothermic group. This was attenuated by local hypothermia, where $\mu HbO_2$ was reduced from 74±3% to 52±4%. Local perfusion, oral microcirculation and macrocirculatory variables were not affected. Selective local hypothermia improves gastric $\mu HbO_2$ during hemorrhagic shock without relevant side effects. In contrast to systemic hypothermia, regional mucosal hypothermia did not affect perfusion and oxygen supply during hemorrhage. Thus, the increased $\mu HbO_2$ during local hypothermia rather indicates reduced mucosal oxygen demand.

## Introduction

Acute blood loss and ensuing circulatory shock after trauma continues to be a major cause of death. Every year, an estimated 5 million people die worldwide as a result of severe injuries [1]. With the exception of the traumatic event itself, exsanguination is the most frequent cause of immediate death followed in the first 24 hours by CNS injury [2]. Later on, multi organ dysfunction syndrome (MODS) is the leading cause of death among patients who die in intensive

**Funding:** This work was supported by a grant of the Strategic Research Fund, Heinrich-Heine-University (No. 1229) to CV. We acknowledge support by the Heinrich-Heine-University Duesseldorf to cover the publication fee. The funders had no role in study design, data collection and analysis, decision to publish, or preparation of the manuscript.

**Competing interests:** The authors have declared that no competing interests exist.

care units [3]. The ischemic gut seems to be an important factor in the pathogenesis of MODS after traumatic hemorrhagic shock. Despite its function for nutrition absorption, the gastrointestinal system is known to function as a barrier against endotoxins and bacteria residing inside the intestinal lumen. In pathological conditions with reduced splanchnic perfusion, such as hemorrhagic shock, bacteria and gut ischemia provoke an intestinal inflammatory response which leads to dysfunction of additional organs [4]. As gut ischemia seems to be the predominant link to translate splanchnic hypoperfusion into an immune-inflammatory event [4], improving gastrointestinal microcirculation is of vital importance. Current strategies to reverse gut dysfunction focus on early enteral nutrition, which has been shown to improve intestinal barrier function, gastrointestinal immunity and resorptive function and to preserve gastrointestinal mucosal architecture [5]. Further approaches investigate immuno-nutrition, protease inhibitors, e.g. tranexamic acid, and direct peritoneal resuscitation [6]. However, these elaborate strategies were not transferred into clinical practice yet.

Mild hypothermia has been shown to improve cellular function in a variety of tissues, especially brain [7,8] and heart [9,10]. Therapeutic hypothermia is the only proven treatment strategy for neuroprotection after resuscitation from out-of-hospital cardiac arrest and can significantly improve long term neurological survival [11]. In 2015, the International Liaison Committee on Resuscitation emphasized its recommendation of target temperature management between 33–36˚C for adults with out-of-hospital cardiac arrest [12]. In previous experimental studies we could show that mild hypothermia is able to improve gastric microcirculatory oxygenation during pathologic conditions such as hemorrhagic shock [13,14] and hypoxia [15]. Regarding hemorrhagic shock in patients the clinical application of systemic therapeutic hypothermia is limited because of systemic complications and detrimental side effects. Hypothermia is a key factor in the lethal triad, consisting of hypothermia, acidosis and coagulopathy, which -unless broken- has a fatal outcome. The coagulation process consists of multiple enzymatic reactions, which function best at a temperature of 37˚C. Thus, hypothermia can cause deleterious effects on coagulopathy in trauma patients [16], and can significantly increase the mortality up to nearly 100% [17]. In patients who suffered cardiac arrest therapeutic hypothermia was associated with significant changes in central hemodynamics as estimated by mean arterial pressure (MAP), heart rate and lactate levels and increased requirements of vasopressors which persisted in the post-hypothermia phase [18]. To limit adverse systemic side effects, new therapy regimens aim at increasing the local concentration of the active therapeutic compound at the desired target location. Recently, we were able to demonstrate favorable effects of topically administered vasoactive drugs like iloprost and nitroglycerin on gastric mucosal oxygenation and perfusion during hemorrhagic shock in dogs without any systemic side effects [19]. Concerning therapeutic hypothermia, local cooling protocols were investigated. In dogs and rats with traumatic and ischemic brain injury, local brain hypothermia combined with decompressive craniectomy lowered intracranial pressure, reduced brain edema formation and neuroinflammation, and improved survival [20]. Regional topical hypothermia improved myocardial function and reduced necrosis in a pig model of myocardial infarction [21].

To date, no data are available on the effects of gastric cooling on local microcirculatory oxygenation and perfusion. The aim of our present study was 1) to establish a cooling unit for selective and rapid gastric and oral mucosal hypothermia and 2) to analyze the effects of selective cooling of the gastric and oral mucosa on local microcirculatory oxygenation and perfusion as well as the systemic side effects during physiological conditions and during mild hemorrhagic shock.

## Materials and methods

### Animals

The data were derived in a cross-over design from repetitive experiments on six dogs (female foxhounds, weighing 28–36 kg) treated in accordance with the NIH guidelines for animal care. Experiments were performed with approval of the Local Animal Care and Use Committee (North Rhine-Westphalia State Agency for Nature, Environment and Consumer Protection, Recklinghausen, Germany; ref. 84–02.04.2012.A152).

A slightly modified, well-established canine model of hemorrhagic shock was used as published previously [22]. All animals were bred for experimental purposes and obtained from the animal research facility (ZETT, Zentrale Einrichtung für Tierforschung und wissenschaftliche Tierschutzaufgaben) of the Heinrich-Heine-University Duesseldorf. The animal husbandry took place in accordance with the European Directive 2010/63/EU and the National Animal Welfare Act. All animals were kept in kennel maintenance under the care of a keeper and with access to an outdoor area. The dogs were fed daily with dry food (Deukadog nature food lamb and rice, Deutsche Tiernahrung Cremer, Duesseldorf, Germany) and wet food (Rinti Gourmet Beef, Finnern, Verden, Germany). Prior to the experiments, access to food was withheld for 12 h with water *ad libitum* to ensure complete gastric depletion and to avoid changes in mucosal perfusion and oxygenation due to digestive activity. Each dog underwent every experimental protocol in a randomized order and served as its own control study. The experiments were performed at least 3 weeks apart to prevent carryover effects. The experiments were performed under general anesthesia (induction of anesthesia with 4 mg·kg$^{-1}$ propofol, maintenance with sevoflurane, end-tidal concentration of 3.0% (1.5 minimum alveolar concentration (MAC) for dogs)). Following endotracheal intubation the dogs were mechanically ventilated (FiO$_2$ = 0.3, VT = 12.5 ml·kg$^{-1}$, a physiological tidal volume for dogs [23]), and the respiratory frequency adjusted to achieve normocapnia (end-expiratory carbon dioxide (etCO$_2$) = 35 mmHg), verified by continuous capnography (Capnomac Ultima, Datex Instrumentarium, Helsinki, Finland). Throughout the experiments, no additional fluid replacement was carried out to avoid volume effects that could influence tissue perfusion and oxygenation. However, after taking each blood sample, normal saline was infused three times the sampling volume to maintain blood volume. Following the intervention, the dogs were extubated as soon as they showed sufficient spontaneous breathing and protective reflexes. The animals remained under direct supervision of the laboratory personnel until complete recovery from anesthesia and first intake of wet food and water. After examination of all puncture sites and a detailed handover, the dogs were given back to their keepers in the animal research facility. No animal was sacrificed during or after the experiments.

### Measurements

**Systemic hemodynamic and oxygenation variables.** The aorta was catheterized via the left carotid artery for continuous measurement of mean arterial pressure (MAP, Gould-Statham pressure transducers P23ID, Elk Grove, IL) and intermittent arterial blood gas analysis (Rapidlab 860, Bayer AG, Germany). Cardiac output (CO) was determined via transpulmonary thermodilution (PiCCO 4.2 non US, PULSION Medical Systems, Munich, Germany) at the end of each intervention. Arterial oxygen content and oxygen delivery (DO$_2$) were calculated subsequently. Heart rate (HR) was continuously measured by electrocardiography (Powerlab, ADInstruments, Castle Hill, Australia).

**Mucosal oxygenation and perfusion.** μHbO$_2$ and μflow of the gastric and oral mucosa were continuously assessed by tissue reflectance spectrophotometry and laser Doppler

flowmetry (O2C, LEA Medizintechnik, Gießen, Germany), respectively, as detailed previously [24]. White light (450–1000 nm) and laser light (820 nm, 30 mW) were transmitted to the tissue of interest via a microlightguide and the reflected light was analyzed. The wavelength-dependent absorption and overall absorption of the applied white light can be used to calculate the percentage of oxygenated hemoglobin ($\mu HbO_2$) and the relative amount of hemoglobin (rHb). Due to the Doppler effect, magnitude and frequency distribution of changes in wavelength are proportional to the number of blood cells multiplied by the measured mean velocity ($\mu velo$) of these cells. This product is proportional to flow and expressed in arbitrary perfusion units (aU). Hence, this method allows assessment and comparison of oxygenation and perfusion of the same region at the same time. Since light is fully absorbed in vessels with a diameter > 100 μm only the microvascular oxygenation of nutritive vessels of the mucosa is measured. The biggest fraction of total blood volume is stored in venous vessels; therefore, mainly postcapillary oxygenation is measured which represents the critical partial pressure of oxygen ($pO_2$) for ischemia. Reading were obtained by placing one flexible lightguide probe in the mouth facing the buccal side of the oral mucosa and a second probe into the stomach via an orogastric silicone tube and positioning it facing the greater curvature. Continuous evaluation of the signal quality throughout the experiments allows verification of the correct position of the probe tip. The $\mu HbO_2$ and μflow values reported are 5 min means (150 spectra, 2 s each) of the respective intervention under steady state conditions. The nontraumatic access to the gastric mucosa allows the determination of mucosal microcirculation without the need of surgical stress. This is particularly desirable regarding the marked alterations that surgical stress exerts on splanchnic circulation. Under these circumstances reflectance spectrophotometry reliably detects even clinically asymptomatic reductions in $\mu HbO_2$ and highly correlates with the morphologic severity and extent of gastric mucosal tissue injury [25].

**Mucosal microcirculation–Videomicroscopy.** Microcirculatory perfusion of the oral mucosa was assessed intermittently by incident dark field (IDF) -imaging (CytoCam, Braedius Medical, Huizen, Netherlands) as described elsewhere [26]. Illumination was provided by light emitting diodes (LED) at a wavelength of 530 nm, the isosbestic point for deoxy- and oxyhemoglobin, and directed towards the oral mucosa. The reflected and scattered light is filtered and makes red blood cells in capillaries visible. All videos were obtained by the same operator. Videos were saved anonymized for blinded analysis. To assess perfusion, a well-established semiquantitative scoring method, the microcirculatory flow index (MFI), was used to characterize microcirculatory flow as "no flow", "intermittent flow", "sluggish flow", and "continuous flow" [27]. To assess density, the total vessel density (TVD), including perfused and non-perfused microvessels, and perfused vessel density (PVD), including perfused microvessels only, were analyzed using the appropriate software (MicroCirculation Analysis software, Braedius Medical, Huizen Netherlands) [28]. The PVD/TVD ratio is used to express the proportion of perfused vessels (PPV). Only vessels with a diameter smaller than 20 μm are included in the analysis. Thus, the PVD represents the functional capillary density (FCD), which is considered to be the main determinant of microcirculatory blood supply [29].

**Intestinal barrier function.** The disaccharide sucrose (D-Sucrose, Carl Roth, Karlsruhe, Germany) was infused into the stomach (1.66 $g \cdot kg^{-1}$ body weight) via an orogastric tube prior to the induction of hemorrhagic shock. Under physiological conditions, sucrose does not pass intact gastrointestinal mucosa. After being transported from the gastric region into the small intestine, ingested sucrose is rapidly degraded by sucrose-isomaltase into monosaccharides. Therefore, intact sucrose cannot be found in blood plasma under physiological conditions. However, under compromising conditions such as shock, barrier function is reduced, and sucrose can pass the gastric mucosal barrier into the plasma, where it does not undergo any enzymatic reaction. Sucrose plasma levels can therefore be used to assess gastric mucosal

barrier function [30]. Blood samples were collected under baseline conditions and during hemorrhagic shock. The collected samples were prepared as previously described [19]. 10 μM ribitol (Adonitol, Carl Roth, Karlsruhe, Germany) was added as internal standard, the samples were dried using a speed vacuum concentrator, measured by gas chromatography-mass spectrometry (GC-MS) and analysed using an appropriate software as described elsewhere [31]. Results are presented as area under the curve (AUC) normalized to the internal standard.

## Induction of local hypothermia

A custom-made large bore orogastric tube with a gastric balloon at its distal tip was inserted into the stomach. The gastric cooling unit consisted of a reservoir with ice cold water, a circulation pump (PA-B1, IKA, Staufen, Germany), gastric balloon, inflow and outflow tube, suitable valves and tubing and a thermometer (GTH 1160, Greisinger electronic, Regenstauf, Germany) attached to the distal tip of the orogastric tube. Cooling of the oral mucosa was achieved by perfusion of a hand-made plastic pad with ice cold water via a syringe pump (Perfusor secura®, Braun, Melsungen, Germany). To maintain normal body temperature, the dogs were covered with isolating blankets and placed on a two-layer mattress (Moek Warming System® Universal III, Moeck & Moeck, Hamburg, Germany) circulated with warm air (Bair Hugger® Model 500, Augustine Medical, Eden Prairie, MN, USA).

## Induction of hemorrhagic shock

Hemorrhagic shock was induced by removing 20% of the estimated total blood volume via a large bore intravenous cannula in a peripheral vein and the arterial catheter (i.e. 16 ml·kg$^{-1}$ of whole blood over five minutes). According to Advanced Trauma Life Support this model represents a class II shock (blood loss 15–30%) [32]. This reversible and nonlethal shock model allows the investigation of either protective or harmful effects of various interventions. The extracted blood was heparinized, stored and later retransfused using an infusion set with a 200 μm filter.

## Experimental protocol

After instrumentation, 30 min were allowed to establish steady state conditions and baseline values were recorded before the animals were randomized to the respective protocol (Fig 1). Steady state conditions were defined as stability of hemodynamic variables (heart rate, mean arterial pressure) as well as ventilation parameters (end-tidal $CO_2$, end-tidal sevoflurane concentration, inspiratory oxygen fraction).

**Local hypothermia + normovolemia (HT-N).**   To study the effects of local hypothermia, after 30 min stable baseline conditions, cooling of the gastric and oral mucosa was initiated by perfusion of the gastric and oral cooling unit with ice cold water. The flow rate was adjusted to achieve a constant mucosal temperature of 34 ± 0.3˚C. All variables were recorded for the next 2.5 h.

**Normothermia + normovolemia (NT-N).**   As time control experiment, gastric and oral mucosa temperatures were kept at 37.5 ± 0.3˚C by flushing with warm water and all variables were recorded for 2.5 h.

**Local hypothermia + hemorrhagic shock (HT-H).**   To study the effect of local hypothermia during hemorrhage and retransfusion, gastric and oral mucosa were cooled as described above and values were recorded for 30 min. Then, hemorrhagic shock was induced and maintained for 60 min, followed by retransfusion of the shed blood with an additional observational period of 60 min.

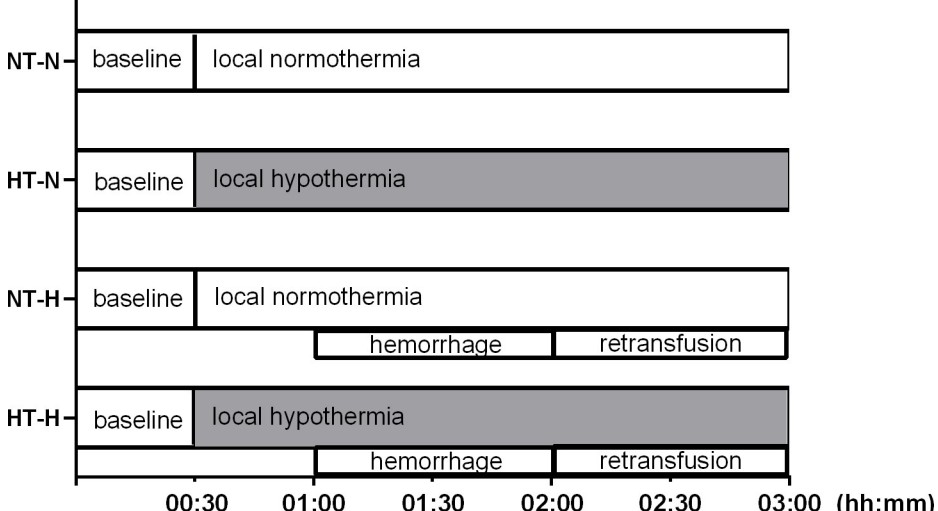

**Fig 1. Experimental protocol.** Experimental protocol: normothermia during normovolemia (NT-N), local gastric and oral hypothermia during normovolemia (HT-N), normothermia during mild hemorrhagic shock (NT-H), and local gastric and oral hypothermia during mild hemorrhagic shock (HT-H).

**Normothermia + hemorrhagic shock (NT-H).** As time control experiment, gastric and oral mucosa temperatures were adjusted to 37.5 ± 0.3˚C by flushing with warm water. Then, hemorrhagic shock was induced and maintained for 60 min followed by retransfusion of the shed blood with an additional observational period of 60 min.

At the end of each intervention, blood samples were obtained for blood gas analysis.

## Statistical analysis

Data for analysis were obtained during the last 5 min of baseline and intervention periods under steady-state conditions. All data are presented as absolute values of mean ± standard error (mean ± SEM) for six dogs. Differences within the groups and between the groups were tested using a two-way analysis of variance for repeated measurements (ANOVA) and a Bonferroni test as post hoc test (GraphPad Prism version 6.05 for Windows, GraphPad Software, La Jolla California USA). An *a priori* power analysis (G*Power Version 3.1.9.2) [33] revealed a power of 0.85 for detection of differences between the different groups with n = 6 in 4 groups, repeated measurements, α < 0.05 and $\eta^2$ of 0.5 (calculated from previous experiments).

## Results

### Temperature management

Both the oral and gastric mucosa were rapidly cooled down in the non-shock group (HT-N) from 37.7 ± 0.2˚C to 33.9 ± 0.2˚C (oral mucosa, S1 Table) and from 38.1 ± 0.3˚C to 34.0 ± 0.1˚C (gastric mucosa, Fig 2) respectively, whereas the mucosal temperature remained normothermic in the control group (NT-N). In the hemorrhagic shock group (HT-H) local cooling reduced oral mucosal temperature from 37.9 ± 0.3˚C to 33.8 ± 0.1˚C (S1 Table) and gastric temperature from 38.3 ± 0.2˚C to 34.0 ± 0.0˚C (Fig 2). In the normothermic control group mucosal temperature remained unchanged throughout the experiment (NT-H). No relevant changes in blood temperature were observed in the normothermic control groups during the experiment. Regional cooling of the oral and gastric mucosa led to a slight decline in blood temperature from 38.2 ± 0.1˚C to 37.2 ± 0.1˚C in the non-shock group (HT-N) and

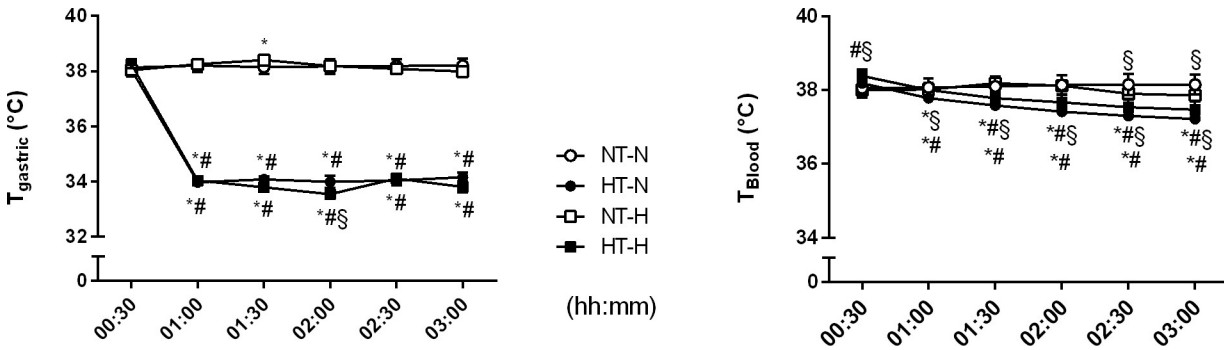

**Fig 2. Temperature of the gastric mucosa and body core temperature.** Gastric mucosal temperature and blood temperature in dogs randomized to normothermia during normovolemia (NT-N), local hypothermia during normovolemia (HT-N), normothermia during mild hemorrhagic shock (NT-H), and local hypothermia during mild hemorrhagic shock (HT-H). Data are presented as mean ± SEM for n = 6 dogs; * = p < 0.05 vs. baseline, # = p < 0.05 vs. respective normothermic control group during physiological conditions (HT-N vs. NT-N) and hemorrhagic shock (HT-H vs. NT-H), § = p < 0.05 vs. respective normovolemic control group during normothermic conditions (NT-H vs. NT-N) and hypothermia (HT-H vs. HT-N), 2-way ANOVA for repeated measurements followed by Bonferroni post hoc test.

from 38.4 ± 0.2˚C to 37.5 ± 0.2˚C in the shock group (HT-H) (Fig 2). In the first 30 min after initiation of local mucosal cooling body core temperature decreased by 0.4 ± 0.1˚C in groups HT-N and HT-H.

## Effect of local hypothermia under physiological conditions (HT-N vs. NT-N)

**Gastric microcirculation.** Under physiological conditions, 2.5 h after onset of mucosal cooling gastric µHbO$_2$ was higher (78 ± 4%) compared to the normothermic control group (69 ± 3%) (Fig 3). Local gastric hypothermia raised gastric µflow from 135 ± 27 aU to 228 ± 43 aU and gastric µvelo from 18 ± 2 aU to 23 ± 4 aU, however without differences to the control group (µflow: 113 ± 20 aU to 149 ± 24 aU; µvelo: 15 ± 1 aU to 18 ± 1 aU) (Table 1).

**Oral microcirculation.** Targeted mucosal cooling did not influence local oral microcirculatory variables. The oral vascular density parameters TVD (Fig 4) and PVD as well as the perfusion parameters MFI (Fig 4) and PPV remained unchanged after initiation of local cooling during otherwise physiological conditions (Table 1).

**Global hemodynamics.** Cooling restricted to the oral and gastric mucosa had no effect on systemic hemodynamic variables. MAP, DO$_2$, CO and SVR remained stable throughout the experiment without differences between the normothermic and hypothermic groups (Table 2).

## Effect of local hypothermia under hemorrhagic conditions (HT-H vs. NT-H)

**Gastric microcirculation.** In the normothermic control group (group NT-H) gastric µHbO$_2$ was strongly reduced during acute hemorrhage (decrease from 72 ± 2% to 38 ± 3%). Cooling of the gastric mucosa (group HT-H) ameliorated the shock induced decrease in mucosal oxygenation and led to an attenuated reduction in µHbO$_2$ from 74 ± 3% to 52 ± 4% and thus remained higher under hypothermia than under normothermia (p < 0.05 vs. NT-H, Fig 3). Local hypothermia did not alter mucosal perfusion variables compared to the normothermic group during hemorrhage (Table 1).

**Oral microcirculation.** Hemorrhagic shock led to a pronounced reduction in oral µHbO$_2$ in the control group from 82 ± 2% to 36 ± 5%. In contrast to gastric microcirculation,

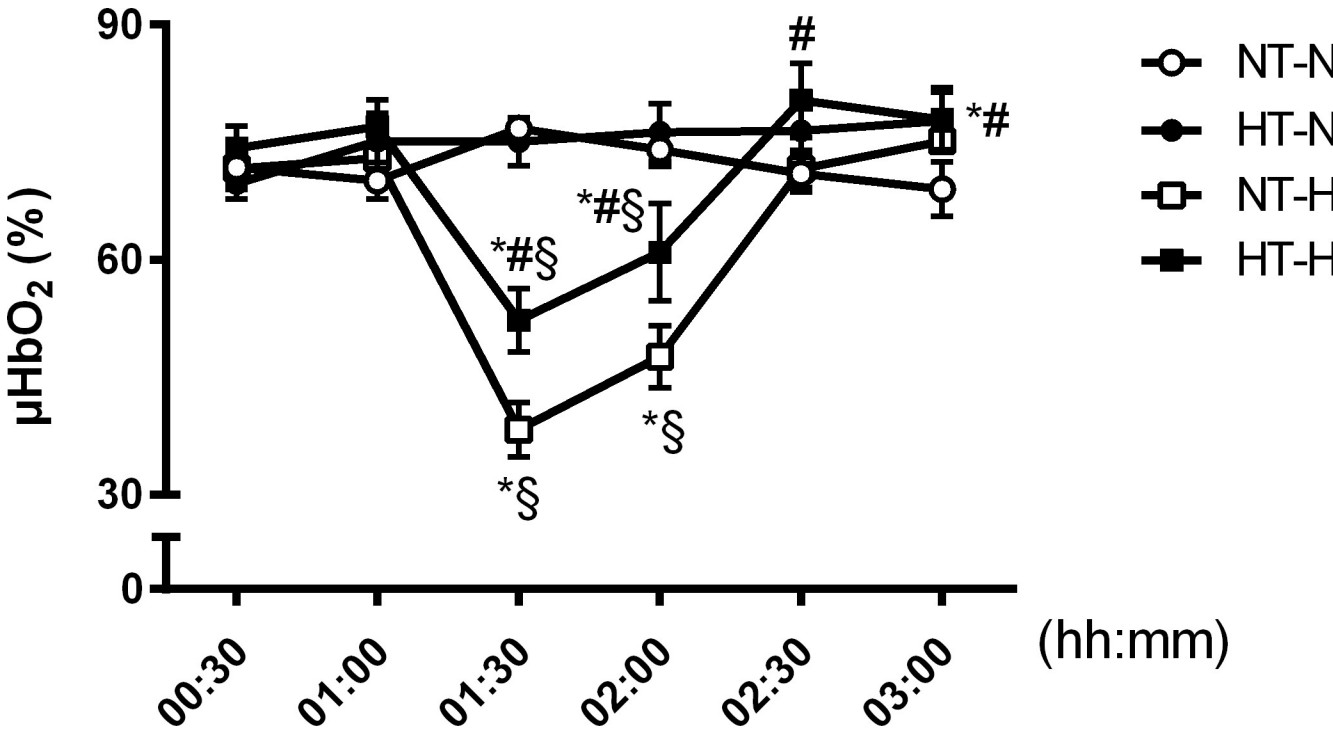

**Fig 3. Gastric μHbO₂.** Gastric microcirculatory oxygenation in dogs randomized to normothermia during normovolemia (NT-N), local hypothermia during normovolemia (HT-N), normothermia during mild hemorrhagic shock (NT-H), and local hypothermia during mild hemorrhagic shock (HT-H). Data are presented as mean ± SEM for n = 6 dogs; * = p < 0.05 vs. baseline, # = p < 0.05 vs. respective normothermic control group during physiological conditions (HT-N vs. NT-N) and hemorrhagic shock (HT-H vs. NT-H), § = p < 0.05 vs. respective normovolemic control group during normothermic conditions (NT-H vs. NT-N) and hypothermia (HT-H vs. HT-N), 2-way ANOVA for repeated measurements followed by Bonferroni post hoc test.

oral μHbO₂ decreased in the hypothermic intervention group from 79 ± 2% to 41 ± 6% without differences between the groups. Likewise, oral μflow decreased from 127 ± 17 aU to 66 ± 14 aU (group NT-H) and from 150 ± 21 aU to 101 ± 27 aU (group HT-H) without differences between the groups (Table 1). Oral buccal capillary density (TVD) measured with IDF was reduced in hemorrhagic shock (12.3 ± 1.6 mm / mm²) compared to baseline values (17.5 ± 1.6 mm / mm²). In contrast, TVD remained stable during hemorrhage (15.2 ± 1.3 mm / mm²) compared to baseline values (17.0 ± 0.8 mm / mm²) when exposed to local hypothermia (Fig 4). However, mucosal cooling did not improve overall capillary perfusion (PVD, PPV). Hemorrhagic shock led to a more inhomogeneous flow pattern in the normothermic and hypothermic group. The overall strongly reduced flow quality (MFI) during hemorrhagic shock (1.7 ± 0.1) was markedly increased in the hypothermic group (2.0 ± 0.2) (Fig 4).

**Intestinal barrier function.** Plasma levels of sucrose increased during the course of the experiment already under physiologic conditions (NT-N) with a rather large inter-individual variability. Hence, the observed differences within and between the groups HT-N, NT-H an HT-H failed to reach significance (Table 3).

**Global hemodynamics.** DO₂ decreased during hemorrhage in both groups: from 13 ± 2 ml·kg⁻¹·min⁻¹ to 7 ± 1 ml·kg⁻¹·min⁻¹ (NT-H) and from 12 ± 1 ml·kg⁻¹·min⁻¹ to 7 ± 1 ml·kg⁻¹·min⁻¹ (HT-H). The decrease of DO₂ is based on a similar decrease in cardiac output. After 30 min of retransfusion DO₂ was restored similarly to 12 ± 1 ml·kg⁻¹·min⁻¹ (NT-H) and 12 ± 1 ml·kg⁻¹·min⁻¹ (HT-H) without differences between both groups. After a 60 min retransfusion period DO₂ was significantly higher in the normothermic group (13 ± 2 ml·kg⁻¹·min⁻¹) compared to the hypothermia group (12 ± 1 ml·kg⁻¹·min⁻¹). MAP,

**Table 1. Microcirculatory variables.**

| Parameter [hh:mm] | Group | 00:30 | | | | 01:00 | | | | 01:30 | | | | 02:00 | | | | 02:30 | | | | 03:00 | | | |
|---|---|---|---|---|---|---|---|---|---|---|---|---|---|---|---|---|---|---|---|---|---|---|---|---|---|
| gastric μflow [aU] | NT-N | 113 | ± | 20 | | 130 | ± | 22 | | 149 | ± | 24 | | 147 | ± | 31 | | 157 | ± | 46 | | 156 | ± | 43 | |
| | HT-N | 135 | ± | 27 | | 174 | ± | 22 | | 228 | ± | 43 | * | 191 | ± | 26 | * | 159 | ± | 22 | | 176 | ± | 35 | |
| | NT-H | 127 | ± | 32 | | 133 | ± | 26 | | 112 | ± | 27 | | 126 | ± | 29 | | 167 | ± | 33 | | 180 | ± | 23 | |
| | HT-H | 148 | ± | 14 | | 185 | ± | 14 | | 140 | ± | 21 | | 175 | ± | 20 | | 212 | ± | 23 | * | 185 | ± | 18 | |
| gastric μvelo [aU] | NT-N | 15 | ± | 1 | | 17 | ± | 1 | | 18 | ± | 1 | | 19 | ± | 2 | | 19 | ± | 3 | | 19 | ± | 3 | |
| | HT-N | 18 | ± | 2 | | 19 | ± | 2 | | 23 | ± | 4 | * | 20 | ± | 2 | | 18 | ± | 2 | | 20 | ± | 3 | |
| | NT-H | 14 | ± | 1 | | 17 | ± | 2 | | 15 | ± | 2 | | 16 | ± | 2 | | 19 | ± | 2 | * | 20 | ± | 2 | * |
| | HT-H | 15 | ± | 1 | | 20 | ± | 1 | * | 17 | ± | 1 | | 19 | ± | 1 | | 22 | ± | 2 | * | 21 | ± | 1 | * |
| gastric rHb [aU] | NT-N | 46 | ± | 4 | | 46 | ± | 7 | | 45 | ± | 6 | | 47 | ± | 6 | | 47 | ± | 5 | | 48 | ± | 5 | |
| | HT-N | 49 | ± | 6 | | 50 | ± | 4 | | 59 | ± | 5 | | 52 | ± | 7 | | 52 | ± | 6 | | 47 | ± | 6 | |
| | NT-H | 50 | ± | 6 | | 48 | ± | 7 | | 40 | ± | 5 | | 40 | ± | 5 | | 48 | ± | 6 | | 49 | ± | 7 | |
| | HT-H | 48 | ± | 2 | | 51 | ± | 3 | | 45 | ± | 4 | | 43 | ± | 2 | | 47 | ± | 3 | | 52 | ± | 7 | |
| oral μHbO$_2$ [%] | NT-N | 73 | ± | 1 | | 75 | ± | 1 | | 78 | ± | 3 | | 82 | ± | 2 | * | 82 | ± | 2 | * | 84 | ± | 2 | * |
| | HT-N | 83 | ± | 2 | # | 83 | ± | 2 | # | 80 | ± | 1 | | 80 | ± | 2 | | 83 | ± | 3 | | 83 | ± | 2 | |
| | NT-H | 82 | ± | 2 | § | 76 | ± | 3 | § | 36 | ± | 5 | *§ | 43 | ± | 3 | *§ | 80 | ± | 4 | | 91 | ± | 2 | * |
| | HT-H | 79 | ± | 2 | | 74 | ± | 3 | § | 41 | ± | 6 | *§ | 50 | ± | 5 | *§ | 78 | ± | 5 | | 85 | ± | 3 | |
| oral μflow [aU] | NT-N | 139 | ± | 30 | | 157 | ± | 36 | | 159 | ± | 38 | | 169 | ± | 41 | | 182 | ± | 39 | | 192 | ± | 36 | * |
| | HT-N | 158 | ± | 31 | | 163 | ± | 39 | | 167 | ± | 38 | | 175 | ± | 37 | | 187 | ± | 37 | | 208 | ± | 42 | * |
| | NT-H | 127 | ± | 17 | | 129 | ± | 15 | | 66 | ± | 14 | * | 75 | ± | 19 | * | 160 | ± | 31 | | 248 | ± | 41 | * |
| | HT-H | 150 | ± | 21 | | 144 | ± | 24 | | 101 | ± | 27 | * | 116 | ± | 30 | | 158 | ± | 37 | | 162 | ± | 34 | |
| oral μvelo [aU] | NT-N | 34 | ± | 6 | | 22 | ± | 3 | | 34 | ± | 7 | | 31 | ± | 7 | | 29 | ± | 5 | | 36 | ± | 5 | |
| | HT-N | 31 | ± | 5 | | 29 | ± | 6 | | 32 | ± | 7 | | 28 | ± | 3 | | 36 | ± | 7 | | 31 | ± | 7 | |
| | NT-H | 23 | ± | 3 | | 26 | ± | 3 | | 19 | ± | 3 | | 23 | ± | 6 | | 28 | ± | 4 | | 35 | ± | 4 | |
| | HT-H | 25 | ± | 2 | | 33 | ± | 6 | | 27 | ± | 7 | | 27 | ± | 6 | | 26 | ± | 5 | | 37 | ± | 7 | |
| oral rHb [aU] | NT-N | 93 | ± | 3 | | 93 | ± | 4 | | 87 | ± | 3 | | 86 | ± | 4 | | 93 | ± | 3 | | 96 | ± | 3 | |
| | HT-N | 92 | ± | 3 | | 92 | ± | 3 | | 87 | ± | 2 | | 89 | ± | 2 | | 89 | ± | 4 | | 87 | ± | 3 | |
| | NT-H | 93 | ± | 2 | | 91 | ± | 3 | | 66 | ± | 4 | *§ | 67 | ± | 2 | *§ | 90 | ± | 2 | | 98 | ± | 2 | |
| | HT-H | 98 | ± | 2 | | 92 | ± | 1 | | 69 | ± | 2 | *§ | 70 | ± | 3 | *§ | 87 | ± | 4 | * | 95 | ± | 5 | |
| PVD | NT-N | 4.9 | ± | 1.6 | | 5.9 | ± | 1.7 | | 7.6 | ± | 1.9 | | 8.3 | ± | 1.8 | * | 7.1 | ± | 1.7 | | 5.1 | ± | 1.5 | |
| | HT-N | 6.2 | ± | 0.8 | | 7.4 | ± | 1.0 | | 5.7 | ± | 1.2 | | 9.0 | ± | 1.5 | | 7.5 | ± | 1.1 | | 8.9 | ± | 1.5 | |
| | NT-H | 6.0 | ± | 1.7 | | 6.0 | ± | 1.3 | | 1.8 | ± | 0.8 | * | 2.9 | ± | 0.9 | | 8.1 | ± | 1.5 | | 8.3 | ± | 1.3 | |
| | HT-H | 6.6 | ± | 1.1 | | 6.9 | ± | 1.2 | | 3.3 | ± | 1.2 | * | 3.9 | ± | 0.9 | | 6.0 | ± | 1.1 | | 8.0 | ± | 1.2 | |
| PPV | NT-N | 27.5 | ± | 7.9 | | 33.2 | ± | 8.5 | | 40.3 | ± | 7.3 | | 41.4 | ± | 7.9 | | 38.5 | ± | 7.9 | | 29.2 | ± | 7.8 | |
| | HT-N | 35.5 | ± | 3.8 | | 42.1 | ± | 5.7 | | 32.6 | ± | 5.9 | | 49.4 | ± | 6.6 | | 42.2 | ± | 4.6 | | 45.8 | ± | 5.9 | |
| | NT-H | 31.6 | ± | 6.4 | | 34.5 | ± | 4.9 | | 11.5 | ± | 4.1 | * | 20.3 | ± | 4.2 | | 42.4 | ± | 6.6 | | 44.4 | ± | 5.5 | |
| | HT-H | 38.1 | ± | 5.0 | | 39.2 | ± | 5.6 | | 20.2 | ± | 6.5 | * | 23.4 | ± | 4.3 | | 32.8 | ± | 5.9 | | 43.7 | ± | 5.3 | |

Microcirculatory variables of the experimental groups–oral mucosal microcirculatory hemoglobin oxygenation (μHbO$_2$), microcirculatory perfusion (μflow, μvelo), relative amount of haemoglobin (rHb) and perfused vessel density (PVD) and proportion of perfused vessels (PPV). Data are presented as absolute values, mean ± SEM, n = 6

* = p < 0.05 vs. baseline

\# = p < 0.05 vs. respective normothermic control group during physiological conditions (HT-N vs. NT-N) and hemorrhagic shock (HT-H vs. NT-H)

§ = p < 0.05 vs. respective normovolemic control group during normothermic conditions (NT-H vs. NT-N) and hypothermia (HT-H vs. HT-N). 2-way ANOVA for repeated measurements followed by Bonferroni post hoc test.

SVR and dPmax were reduced during hemorrhagic shock in both groups without differences between normothermia and hypothermia (Table 2). Local hypothermia had no effect

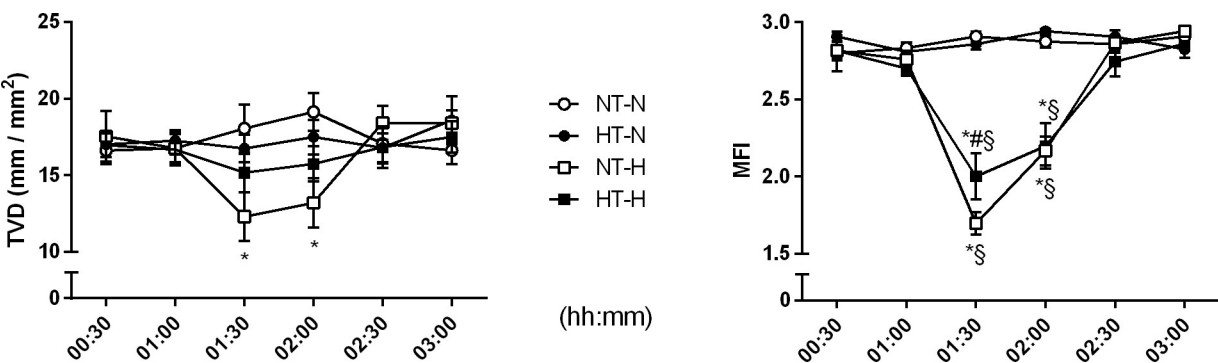

**Fig 4. Oral microvascular vessel density and microvascular flow index.** Total vessel density (TVD) and microvascular flow index (MFI) of the oral mucosa assessed with IDF-imaging in dogs subjected to normothermia during normovolemia (NT-N), local hypothermia during normovolemia (HT-N), normothermia during mild hemorrhagic shock (NT-H), and local hypothermia during mild hemorrhagic shock (HT-H). Data are presented as mean ± SEM for n = 6 dogs; * = p < 0.05 vs. baseline, # = p < 0.05 vs. respective normothermic control group during physiological conditions (vs. NT-N for HT-N) and hemorrhagic shock (vs. NT-H for HT-H), § = p < 0.05 vs. respective normovolemic control group during normothermic conditions (vs. NT-N for NT-H) and hypothermia (vs. HT-N for HT-H), 2-way ANOVA for repeated measurements followed by Bonferroni post hoc test.

on systemic metabolic variables during hemorrhage. Base excess and pH were equally reduced and $pCO_2$ was increased in both groups. During hemorrhagic shock, no clinically relevant alterations in lactate plasma levels were observed (S2 Table).

## Discussion

The aim of our study was to analyze the effects of local gastric and oral mucosal mild hypothermia on regional microcirculation and systemic hemodynamic variables during physiologic conditions and during a mild hemorrhagic shock.

The main findings are:

1. Local mucosal hypothermia (34˚C) attenuates the decrease in gastric microcirculatory oxygenation during hemorrhagic shock.

2. Locally induced hypothermia enhances gastric mucosal perfusion during normovolemic conditions but not during subsequent hemorrhagic shock.

3. Local cooling increases oral capillary total vessel density and shifts capillary perfusion to a more continuous flow pattern during hemorrhagic shock but fails to enhance the overall microcirculatory perfusion.

4. Cooling of the oral and gastric mucosa to 34˚C does not influence systemic hemodynamic variables, neither during physiologic conditions nor in hemorrhagic shock.

This study shows that local cooling of the gastric mucosa is able to notably attenuate the shock induced decrease in microcirculatory oxygenation. These results are in accordance with our previous results demonstrating improved gastric $\mu HbO_2$ by systemic hypothermia during hemorrhagic shock [13,14] and hypoxia [15]. There are several possible reasons for the observed increase in gastric $\mu HbO_2$. The increase in microcirculatory oxygenation might be mediated either by an increased oxygen supply or by a reduction of oxygen consumption. The reduced oxygen consumption could be related to a reduced oxygen demand or to the inability to extract oxygen. The measurement of $\mu$flow in our experiments demonstrates that the observed increase of $\mu HbO_2$ is not based on an enhanced microcirculatory perfusion coupled with an increase in regional oxygen delivery. Thus, the improved oxygenation during

**Table 2. Macrohemodynamic variables.**

| Parameter [hh:mm] | Group | 00:30 | 01:00 | 01:30 | 02:00 | 02:30 | 03:00 |
|---|---|---|---|---|---|---|---|
| $DO_2$ [ml·kg$^{-1}$·min$^{-1}$] | NT-N | 11 ± 1 | 11 ± 1 | 12 ± 2 | 12 ± 2 | 12 ± 2 | 12 ± 2 |
| | HT-N | 12 ± 2 | 12 ± 1 | 12 ± 2 | 12 ± 2 | 11 ± 2 | 12 ± 1 |
| | NT-H | 13 ± 2 | 12 ± 2 | 7 ± 1 §* | 9 ± 1 §* | 12 ± 1 | 13 ± 2 § |
| | HT-H | 12 ± 1 | 12 ± 1 | 7 ± 1 §* | 8 ± 1 §* | 12 ± 1 | 12 ± 1 # |
| SVR [mmHg·l$^{-1}$·min$^{-1}$] | NT-N | 31 ± 2 | 31 ± 2 | 31 ± 2 | 30 ± 2 | 30 ± 2 | 30 ± 2 |
| | HT-N | 27 ± 3 # | 29 ± 3 | 29 ± 3 | 29 ± 3 | 30 ± 3 | 28 ± 3 |
| | NT-H | 27 ± 3 § | 29 ± 2 | 37 ± 3 §* | 36 ± 4 §* | 31 ± 2 * | 27 ± 2 |
| | HT-H | 28 ± 3 | 29 ± 3 | 40 ± 5 §* | 40 ± 5 §* | 33 ± 2 * | 29 ± 3 |
| CO [ml·kg$^{-1}$·min$^{-1}$] | NT-N | 74 ± 6 | 74 ± 6 | 76 ± 7 | 78 ± 8 | 78 ± 7 | 78 ± 8 |
| | HT-N | 79 ± 8 | 75 ± 6 | 76 ± 7 | 76 ± 7 | 74 ± 8 | 77 ± 6 |
| | NT-H | 81 ± 8 § | 78 ± 6 | 47 ± 3 *§ | 56 ± 6 *§ | 81 ± 6 § | 86 ± 7 |
| | HT-H | 80 ± 7 | 77 ± 7 | 44 ± 5 *§ | 50 ± 4 *§ | 75 ± 4 | 79 ± 5 |
| SV [ml] | NT-N | 19 ± 2 | 19 ± 2 | 20 ± 2 | 20 ± 2 | 21 ± 2 * | 21 ± 2 * |
| | HT-N | 21 ± 3 # | 20 ± 2 | 21 ± 2 | 21 ± 2 | 21 ± 2 | 22 ± 2 |
| | NT-H | 22 ± 3 § | 21 ± 2 | 13 ± 2 §* | 15 ± 2 §* | 24 ± 3 §* | 25 ± 3 §* |
| | HT-H | 21 ± 2 | 20 ± 2 | 12 ± 2 §* | 13 ± 2 §* | 23 ± 2 § | 24 ± 2 * |
| MAP [mmHg] | NT-N | 68 ± 3 | 69 ± 2 | 69 ± 3 | 69 ± 3 | 69 ± 3 | 69 ± 3 |
| | HT-N | 62 ± 2 | 65 ± 2 | 65 ± 2 | 65 ± 2 | 65 ± 2 | 64 ± 2 |
| | NT-H | 64 ± 2 | 68 ± 3 * | 52 ± 2 * | 60 ± 2 | 76 ± 4 * | 69 ± 3 * |
| | HT-H | 65 ± 2 | 66 ± 2 | 52 ± 2 * | 59 ± 2 * | 76 ± 3 * | 69 ± 2 * |
| HR [min$^{-1}$] | NT-N | 121 ± 4 | 119 ± 4 | 117 ± 4 | 117 ± 4 | 115 ± 4 * | 114 ± 4 * |
| | HT-N | 117 ± 3 | 117 ± 4 | 116 ± 4 | 114 ± 4 | 112 ± 4 | 111 ± 3 * |
| | NT-H | 119 ± 4 | 119 ± 3 | 116 ± 5 | 121 ± 5 | 105 ± 3 * | 109 ± 3 * |
| | HT-H | 118 ± 3 | 118 ± 4 | 114 ± 5 | 119 ± 4 | 102 ± 2 * | 104 ± 2 * |
| dPmax [mmHg·min$^{-1}$] | NT-N | 267 ± 14 | 262 ± 18 | 268 ± 20 | 277 ± 22 | 283 ± 22 | 282 ± 20 |
| | HT-N | 300 ± 24 | 273 ± 24 | 270 ± 24 | 272 ± 22 | 271 ± 24 | 282 ± 18 |
| | NT-H | 298 ± 22 | 277 ± 22 | 248 ± 14 * | 263 ± 19 * | 303 ± 27 | 310 ± 30 |
| | HT-H | 305 ± 33 | 278 ± 26 | 234 ± 11 §* | 252 ± 13 * | 285 ± 19 | 302 ± 24 |

Systemic hemodynamic variables of the experimental groups–mean arterial pressure (MAP), heart rate (HR), systemic oxygen delivery ($DO_2$), cardiac output (CO) and systemic vascular resistance (SVR). Data are presented as absolute values, mean ± SEM, n = 6

* = $p < 0.05$ vs. baseline

# = $p < 0.05$ vs. respective normothermic control group during physiological conditions (HT-N vs. NT-N) and hemorrhagic shock (HT-H vs. NT-H)

§ = $p < 0.05$ vs. respective normovolemic control group during normothermic conditions (NT-H vs. NT-N) and hypothermia (HT-H vs. HT-N). 2-way ANOVA for repeated measurements followed by Bonferroni post hoc test.

hemorrhage in this study might rather be caused by a modulated oxygen consumption. These results are in contrast with our previous findings that systemic hypothermia improved gastric μflow during hemorrhagic shock [13]. Mild hypothermia in dogs might shift the oxygen-hemoglobin-dissociation curve to the left, thereby increasing the affinity of oxygen to hemoglobin [34]. However, in hypothermic dogs with a body core temperature of 30.5 ± 0.1˚C no evidence was found that a left ward shift of the oxygen-hemoglobin-dissociation curve impaired the transport of $O_2$ into the peripheral tissue [35]. In cardiac surgical patients moderate hypothermia of 32˚C during extracorporeal circulation did not seem to induce changes in peripheral tissue oxygen delivery [36]. Thus, increased hemoglobin binding affinity for oxygen with consecutive reduced oxygen delivery does not sufficiently explain the increase in capillary

**Table 3. Sucrose plasma concentrations.**

| Parameter [hh:mm] | Group | 00:30 | | | | 01:00 | | | | 01:30 | | | | 02:00 | | | |
|---|---|---|---|---|---|---|---|---|---|---|---|---|---|---|---|---|---|
| Sucrose [rel. amount / µl plasma] | NT-N | 2.4 | ± | 1.2 | | 109.1 | ± | 63.6 | * | 175.9 | ± | 85.2 | *# | 166.9 | ± | 64.9 | * |
| | HT-N | 4.9 | ± | 3.3 | | 28.6 | ± | 14.7 | | 67.6 | ± | 28.9 | | 68.0 | ± | 26.1 | |
| | NT-H | 2.4 | ± | 1.0 | | 31.3 | ± | 20.3 | | 96.1 | ± | 55.9 | | 94.4 | ± | 48.5 | |
| | HT-H | 22.0 | ± | 19.6 | | 14.0 | ± | 3.7 | | 78.6 | ± | 25.7 | | 79.8 | ± | 20.9 | |

Sucrose plasma levels. Data are presented as absolute values, mean ± SEM, n = 6

* = $p < 0.05$ vs. baseline

# = $p < 0.05$ vs. respective normothermic control group during physiological conditions (HT-N vs. NT-N) and hemorrhagic shock (HT-H vs. NT-H). 2-way ANOVA for repeated measurements followed by Bonferroni post hoc test.

and postcapillary $\mu HbO_2$. On the other hand hypothermia is known to decrease the loss in cellular high-energy phosphates, slow down the rates of metabolites consumption and lactic acid accumulation [37] and to reduce the cerebral metabolic rate by 6% for every 1˚C reduction in brain temperature [38]. Mild and moderate hypothermia protects enterocyte mitochondrial function during hemorrhagic shock, reduces intestinal injury and apoptosis, modulates inflammatory pathways and increases survival times [39]. Therefore, the assumption seems feasible that the increased $\mu HbO_2$ is mainly attributable to a decrease in local mucosal oxygen consumption, probably due to reduced oxygen demand. As we studied a non-lethal, non-invasive shock model, the exact regional oxygen consumption of the gastric mucosal tissue could not be measured directly. To sum up, in the previous study systemic hypothermia caused an enhanced microcirculatory oxygenation during hemorrhagic shock most likely via the sympathetic nerve system and a vasopressin mediated increase in regional perfusion [13,14]. However, local cooling, as performed in the present study, evokes its effects on regional oxygenation most likely via a reduced local oxygen consumption.

The measurement of sucrose plasma levels did not reveal consistent results. Varying levels of sucrose absorption have been shown to reliably indicate changes in gastric mucosal barrier function[40] and to indicate shock induced mucosal leakage[19]. In the present study, the gastric balloon used for regional cooling likely had a significant impact on the distribution of the sucrose solution. Paired with a highly variable gastric reflux, a steady allocation and uniform gastric resorption of sucrose seems rather unlikely.

During normovolemic conditions, local hypothermia increased gastric µflow in anesthetized dogs compared to normothermic controls. The increase of µflow might be attributed to higher µvelo. Increased µflow and µvelo most likely occur in the presence of regional vasoconstriction. A higher capillary density is unlikely as rHb, a parameter correlating with the amount of blood in the tissues thereby indicating the filling of microvessels, remained unchanged. Hypothermia-induced vasoconstriction is widely reported in various tissues, e.g. skin[41] and bone[42], but the effects of hypothermia on vascular tone seem to be tissue dependent, as cooling of an isolated carotid artery preparation induces a reversible graded vasodilation[43]. During pathologic conditions, regulation of microvascular perfusion is differently affected by hypothermia. Kalia et al. reported that no apparent reduction in intestinal mucosal blood flow was observed in rats subjected to superior mesenteric artery ischemia / reperfusion and subsequent hypothermia[44]. This supports our finding, that gastric µflow is modulated by hypothermia during normovolemic conditions but not in hemorrhagic shock.

Effects of mild local hypothermia on microcirculatory oxygenation during hemorrhage were only observed at the gastric mucosa, whereas oral $\mu HbO_2$ remained similar to the control

group. Thus, oral microcirculation does not seem to be an indicator of gastrointestinal micro-circulation *per se*. During hemorrhagic shock, local cooling led to an increase in TVD and MFI. The MFI represents the quality of microvascular blood flow. However, it does not provide any information on the density of perfused vessels or whether an intervention is able to recruit capillaries [27]. Local cooling did neither lead to an increase in overall oral microvascular perfusion (μflow) nor to a recruitment of underperfused vessels (PVD, PPV). Therefore, it is questionable, whether the improved quality of flow really results in an improved organ perfusion. Still, it is not yet clear which microvascular parameter is the most appropriate to predict organ blood and oxygen supply.

During local cooling, no marked changes were observed in physiological parameters including cardiac output, heart rate, mean arterial pressure, and other metabolic variables. Stable macrohemodynamic parameters were reported previously by Clark et al. [45], during selective brain hypothermia in a rat model of ischemic brain injury. Hypothermia is known to reduce cardiac output [46], probably by reducing cardiac contractility [47] and heart rate [14]. During the chosen local cooling regime, no differences in dPmax, as an indicator for cardiac contractility, and in cardiac output were recorded. The present study documents no differences in heart rate between normothermic and hypothermic animals. Surprisingly, hemorrhagic shock did not result in an increase in heart rate. This could, at least partially be explained by sevoflurane-induced depression of the baroreflex [48]. This phenomenon has been observed in this animal model under hemorrhagic shock before and might be related to the rather mild shock [13,14,19].

Our study has several limitations. The sample size of n = 6 animals per group seems to be rather small compared to usual sample sizes in non-repetitive experiments, e. g. using rats. However, the cross-over design allows sufficient power for small sample sizes, as each animal serves as its own control. The impact of hemorrhage is moderate, as this is a non-lethal animal model. We chose a pretreatment protocol which, despite its limitations, has several advantages. Insertion and prefilling of the gastric cooling unit allows rapid induction of the local hypothermia without disturbing the sensitive measurement of regional microcirculation. Manipulating the cooling balloon during hemorrhagic shock might substantially interfere with the measurement. The beneficial effects of systemic mild hypothermia on gastric microcirculation were seen in a pretreatment protocol. Therefore, a similar pretreatment protocol allows a more valid comparison with these existing data.

## Conclusions

Studies on the effects of hypothermia on gastrointestinal microcirculation are sparse [39,44]. As described above, research is currently being carried out on local cooling protocols in the context of myocardial and neurological injury. In the 1960s local gastric freezing, using gastric cooling units comparable to the one used in the present study, was examined as a potential treatment option for gastrointestinal bleeding [49], but were abandoned due to local mucosal injury and ulcer formation. Implementing a gastric cooling unit close to the large abdominal vessels led to a slight but continuous reduction in core body temperature during the experiment. Although not consistently reported [50], even a minor decline in body temperature can increase intraoperative blood loss [51] and might limit the clinical applicability of local hypothermia. However, mild local cooling of the gastrointestinal mucosa without hemodynamic adverse side effects might be a promising approach that can be applied to the clinical setting of patients with disturbed circulation after gastrointestinal surgery. Important factors preventing anastomotic leak, apart from surgical suture, are sufficient local oxygenation and perfusion

[52] and local mild hypothermia at the side of the anastomosis might support anastomotic healing.

## Supporting information

**S1 Table. Temperature management.** Oral mucosal temperature in all experimental groups. Data are presented as absolute values, mean ± SEM, n = 6, * = $p < 0.05$ vs. baseline, # = $p < 0.05$ vs. respective normothermic control group during physiological conditions (HT-N vs. NT-N) and hemorrhagic shock (HT-H vs. NT-H), § = $p < 0.05$ vs. respective normovolemic control group during normothermic conditions (NT-H vs. NT-N) and hypothermia (HT-H vs. HT-N). 2-way ANOVA for repeated measurements followed by Bonferroni post hoc test.
(DOCX)

**S2 Table. Metabolic variables.** Arterial oxygen partial pressure ($P_aO_2$), carbon dioxide partial pressure ($P_aCO_2$), haematocrit (Hct), pH, bicarbonate ($HCO_3^-$), and lactate plasma levels. Data are presented as absolute values, mean ± SEM, n = 6, * = $p < 0.05$ vs. baseline, # = $p < 0.05$ vs. respective normothermic control group during physiological conditions (HT-N vs. NT-N) and hemorrhagic shock (HT-H vs. NT-H), § = $p < 0.05$ vs. respective normovolemic control group during normothermic conditions (NT-H vs. NT-N) and hypothermia (HT-H vs. HT-N). 2-way ANOVA for repeated measurements followed by Bonferroni post hoc test.
(DOCX)

## Author Contributions

**Conceptualization:** Richard Truse, Christian Vollmer.

**Formal analysis:** Richard Truse, Michael Smyk, Tabea Mettler-Altmann, Christian Vollmer.

**Funding acquisition:** Richard Truse, Inge Bauer, Olaf Picker, Christian Vollmer.

**Investigation:** Richard Truse, Michael Smyk, Jan Schulz, Anna Herminghaus, Christian Vollmer.

**Methodology:** Richard Truse, Andreas P. M. Weber, Tabea Mettler-Altmann, Inge Bauer, Olaf Picker, Christian Vollmer.

**Project administration:** Andreas P. M. Weber, Tabea Mettler-Altmann, Inge Bauer, Olaf Picker.

**Resources:** Andreas P. M. Weber, Inge Bauer, Olaf Picker.

**Supervision:** Inge Bauer, Olaf Picker.

**Validation:** Jan Schulz, Anna Herminghaus, Andreas P. M. Weber, Tabea Mettler-Altmann, Inge Bauer, Olaf Picker, Christian Vollmer.

**Visualization:** Richard Truse, Michael Smyk.

**Writing – original draft:** Richard Truse.

**Writing – review & editing:** Michael Smyk, Jan Schulz, Anna Herminghaus, Andreas P. M. Weber, Tabea Mettler-Altmann, Inge Bauer, Olaf Picker, Christian Vollmer.

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
