## [Decision Letter · Decision Letter 0]

29 Oct 2019

PONE-D-19-24409

Regional hypothermia improves gastric microcirculatory oxygenation during hemorrhage in dogs.

PLOS ONE

Dear Dr. Truse,

Thank you for submitting your manuscript to PLOS ONE. After careful consideration, we feel that it has merit but does not fully meet PLOS ONE’s publication criteria as it currently stands. Therefore, we invite you to submit a revised version of the manuscript that addresses the points raised during the review process.

We would appreciate receiving your revised manuscript by 15.12.2019. To enhance the reproducibility of your results, we recommend that if applicable you deposit your laboratory protocols in protocols.io, where a protocol can be assigned its own identifier (DOI) such that it can be cited independently in the future. For instructions see: http://journals.plos.org/plosone/s/submission-guidelines#loc-laboratory-protocols

We look forward to receiving your revised manuscript.

Kind regards,

Aleksandar R. Zivkovic

Academic Editor

PLOS ONE

2. 

We suggest you thoroughly copyedit your manuscript for language usage, spelling, and grammar. If you do not know anyone who can help you do this, you may wish to consider employing a professional scientific editing service.  

3.  At this time, we request that you  please report additional details in your Methods section regarding animal care, as per our editorial guidelines:

(1) Please state the source of the dogs used in the study  

(2) Please provide details of animal welfare (e.g., shelter, food, water, environmental enrichment)

(3) Please include the method of euthanasia

(4) Please describe in more detail the post-operative care received by the animals, including the frequency of monitoring and the criteria used to assess animal health and well-being. Thank you for your attention to these requests.

Review Comments to the Author

Reviewer #1: It is well designed and precisely performed in an experimental study. The authors demonstrated a beneficial impact of regional hypothermia on maintaining blood gas and acid-base homeostasis and it’s recovering capacity in cases of acute hemorrhagic situations during gastrointestinal surgery or trauma. I recommend accepting after minor revision to calculate sample size and perform power analysis.

Reviewer #2: The authors have developed a regional cooling mechanism to maintain selective local gastric and mucosal mild hypothermia in dogs. Under this protocol, they have analyzed regional microcirculation and systemic hemodynamic variables during a mild hemorrhagic shock. The most conclusive result is that hypothermia preserves mu HbO(2) during hemorrhagic shock, as the same authors verified previously with systemic hypothermia, that does not show remarkable novelty.

Minor points:

The paragraph that begins on line 309, as written, is confusing since it is not focused on the fact that cooling increases oxygenation during hemorrhage.

It is a bit strange to see all the data in a table, and then some repeated in figures, when data do not even offer significant differences.

---

## [Author Response · Author response to Decision Letter 0]

13 Nov 2019

Dear Dr. Zivkovic,

please find attached our point-by-point response to your and the reviewer’s comments: 

We revised the manuscript to meet the journal’s requirements, especially concerning headings, references and file naming.

The manuscript was thoroughly copyedited by an native speaking colleague (Michael Miles Albin Stuebs, Department of Anesthesiology, Duesseldorf University Hospital).

3. At this time, we request that you please report additional details in your Methods section regarding animal care, as per our editorial guidelines:

(1) Please state the source of the dogs used in the study

All animals were bred for experimental purposes and obtained from the animal research facility (ZETT, Zentrale Einrichtung für Tierforschung und wissenschaftliche Tierschutzaufgaben) of the Heinrich-Heine-University Duesseldorf. We have added this information on page 6, lines 101 – 103. 

(2) Please provide details of animal welfare (e.g., shelter, food, water, environmental enrichment)

The animal husbandry took place in accordance with the European Directive 2010/63/EU and the National Animal Welfare Act. All animals were kept in kennel maintenance under the care of a keeper and with access to an outdoor area. The dogs were fed daily with dry food (Deukadog nature food lamb and rice, Deutsche Tiernahrung Cremer, Duesseldorf, Germany) and wet food (Rinti gourmet beef, Finnern, Verden, Germany). Prior to the experiments, access to food was withheld for 12 h with water ad libitum. This information is included in the Method section on page 6, lines 103 – 108.

(3) Please include the method of euthanasia

As we used a non-lethal hemorrhagic shock model, all animals survived, and no animal was euthanized during or after the experiment. On completion of the series of experiments all animals were inspected by the animal welfare officer of the Heinrich-Heine-University and after veterinary evaluation the dogs were utilized in subsequent projects. We emphasized the information that no animal was euthanized during or after the experiments on page 7, lines 127 - 128.

(4) Please describe in more detail the post-operative care received by the animals, including the frequency of monitoring and the criteria used to assess animal health and well-being. Thank you for your attention to these requests.

Following the intervention, the dogs were extubated as soon as they showed sufficient spontaneous breathing and protective reflexes. The animals remained under direct supervision of the laboratory personnel until complete recovery from anesthesia and first intake of wet food and water. After examination of all puncture sites and a detailed handover, the dogs were given back to their keepers in the animal research facility. We included the details on post-operative care in the description of our experiment (page 7, lines 123 – 127). 

Review Comments to the Author

Reviewer #1: It is well designed and precisely performed in an experimental study. The authors demonstrated a beneficial impact of regional hypothermia on maintaining blood gas and acid-base homeostasis and it’s recovering capacity in cases of acute hemorrhagic situations during gastrointestinal surgery or trauma. I recommend accepting after minor revision to calculate sample size and perform power analysis.

The sample size calculation was performed with G*Power Version 3.1.9.2. We performed an a priori analysis for ANOVA with repeated measurements. From previous experiments we calculated a η2 of 0.5. Our protocol with repeated measurements in 4 groups with α < 0.05 reached an actual power of 0.85 with a total sample size of 24, resulting in n=6 per group. We added the results of the power analysis in the methods section / statistical analysis part (page 13, lines 261 – 264).

Reviewer #2: The authors have developed a regional cooling mechanism to maintain selective local gastric and mucosal mild hypothermia in dogs. Under this protocol, they have analyzed regional microcirculation and systemic hemodynamic variables during a mild hemorrhagic shock. The most conclusive result is that hypothermia preserves mu HbO(2) during hemorrhagic shock, as the same authors verified previously with systemic hypothermia, that does not show remarkable novelty.

Indeed, our present study reveals similar effects of local gastric mucosal cooling as we have reported during mild systemic hypothermia previously. However, we pursued a different approach, as we focus on effects of a mild local hypothermia restricted to the organ of interest. This focused approach might limit systemic adverse events that come along with systemic hypothermia and might be clinically applicable under certain circumstances as pointed out in our conclusion section (page 26, lines 516 – 521). Therefore, we are convinced, that our findings contribute to an understanding on tissue protective effects of hypothermia.

Minor points:

The paragraph that begins on line 309, as written, is confusing since it is not focused on the fact that cooling increases oxygenation during hemorrhage.

The paragraph mentioned in this comment is part of the results section of our manuscript. It describes that the mild hemorrhagic shock performed in our experiments led to a pronounced decline in gastric microcirculatory oxygenation. This shock induced decline was significantly ameliorated by prior induction of mild mucosal hypothermia. To enhance the comprehensibility of this paragraph we revised the presentation of these results and added the significance for the comparison of µHbO2 during hemorrhagic shock in the normothermic and hypothermic groups (page 19, lines 353 – 357).

It is a bit strange to see all the data in a table, and then some repeated in figures, when data do not even offer significant differences.

As suggested we revised the figures and tables. The figures display the results of all four experimental groups. Significant differences to baseline values, and the respective normothermic or normovolemic control group are indicated where applicable. All data shown in figures are deleted from the tables. As no significant differences in cardiac output were found between the normothermic and hypothermic group, Figure 5 was removed as suggested.

---

## [Editor Report · Decision Letter 1]

21 Nov 2019

Regional hypothermia improves gastric microcirculatory oxygenation during hemorrhage in dogs.

PONE-D-19-24409R1

Dear Dr. Truse,

We are pleased to inform you that your manuscript has been judged scientifically suitable for publication and will be formally accepted for publication once it complies with all outstanding technical requirements.

With kind regards,

Aleksandar R. Zivkovic

Academic Editor

PLOS ONE

---

## [Editor Report · Acceptance letter]

3 Dec 2019

PONE-D-19-24409R1 

Regional hypothermia improves gastric microcirculatory oxygenation during hemorrhage in dogs. 

Dear Dr. Truse:

I am pleased to inform you that your manuscript has been deemed suitable for publication in PLOS ONE. Congratulations! Your manuscript is now with our production department. 

With kind regards,

on behalf of

Dr. Aleksandar R. Zivkovic 

Academic Editor

PLOS ONE